# Relationship Between Harvesting Efficiency and Filament Morphology in *Arthrospira platensis* Gomont

**DOI:** 10.3390/microorganisms13020367

**Published:** 2025-02-08

**Authors:** Ga-Hyeon Kim, Yeong Jun Lee, Jong-Hee Kwon

**Affiliations:** 1Department of Food Science & Technology, Institute of Agriculture & Life Science, Gyeongsang National University, Jinju 52828, Republic of Korea; kkh56230@naver.com (G.-H.K.); log1124@naver.com (Y.J.L.); 2Division of Applied Life Sciences (BK21), Gyeongsang National University, Jinju 52828, Republic of Korea

**Keywords:** *Arthrospira platensis*, morphology, auto-flotation, photosynthetic activity

## Abstract

*Arthrospira platensis*, a filamentous cyanobacterium, exhibits morphological variability influenced by biotic and abiotic factors. We investigated the effect of sodium ion concentration on filament length, growth, and harvest efficiency. Increasing the sodium concentration from 0.2 M to 0.4 M (using NaHCO_3_ or Na_2_CO_3_) led to a significant increase in filament length, from 0.3393 to 0.7084 mm, and longer filaments had increased auto-flotation efficiency (from 87% to 94%) within 3 h. The linear filaments, obtained via spontaneous morphological conversion, also had increased photosynthetic activity and growth rates compared to coiled filaments, and we speculate this was due to decreased self-shading and increased light penetration. However, linear filaments also had poor auto-flotation efficiency (10% after 24 h) and decreased buoyancy, and this likely limits their survival in natural ecosystems. These findings provide insights into optimizing the cultivation of *A. platensis* for biomass harvesting.

## 1. Introduction

*Spirulina*, whose name comes from a Latin word for “helix” or “spiral” [1,2], is a genus of filamentous multicellular cyanobacteria in the class Cyanophyceae [3]. The dietary supplement named ‘spirulina’ refers to the biomass of a species formerly known as *Spirulina platensis*, but it is now known as *Arthrospira platensis*. *A. platensis* is an aerobic, multicellular, and filamentous photosynthetic cyanobacterium that lives in alkali environments with high levels of salts, such as carbonate and bicarbonate, in subtropical and tropical areas [4,5]. *A. platensis* cells also contain abundant vitamins (e.g., vitamin B_12_ [6], β-carotene [7]), proteins (60–70%) [8], the unsaturated fatty acid gamma-linolenic acid [9], and natural pigments, such as zeaxanthin [10], phycocyanin, and myxoxanthophyll [11]. Dietary supplements prepared from these cells are is utilized to boost the immune system [12,13], increase blood circulation [14,15], and as a food additive [16] and dietary supplement [17,18].

Taxonomic identification of the species in *Arthrospira* is challenging due to morphological variability caused by genetic drift, possibly due to changing environmental conditions. A previous study reported that genomic differences are observed between the helical and linear morphotypes of *Arthrospira* sp. PCC 8005 [19]. In natural ecosystems and cultures, *A. platensis* filaments have two distinct morphological types: spiral or linear [20]. The linearization of *A. platensis* from its spiral form mainly occurs in artificial conditions, such as in laboratories and industrial settings [21], and it is now widely accepted that the linear shape is one of the major morphologies of *A. platensis* [4]. The linear morphology was long considered to be a permanent degeneration that could not be reversed [22], although Wang and Zhao [23] reported that linear filaments can revert to their original spiral morphology under certain conditions. In addition to a permanent morphological degeneration caused by genetic changes, other studies reported alterations in the trichome length and helicity of *A. platensis* in response to external factors, such as changes in light spectrum, salinity, and glucose level [24,25].

Current harvesting methods for *Spirulina* (*A. platensis*) vary based on the scale of production. For laboratory-scale applications, common methods include filtration using membrane filters or mesh sieves [26,27], low-speed centrifugation [28], and sedimentation with or without flocculants [29,30]. In industrial-scale operations, more advanced techniques such as microfiltration [31], industrial centrifugation [32], flocculation-assisted sedimentation [33], belt and rotary drum filtration [34,35], and electrocoagulation [36,37] are employed to efficiently separate biomass while ensuring cost-effectiveness. The choice of harvesting method depends on factors such as scalability, energy consumption, and processing efficiency. Among these methods, auto-flotation offers significant advantages for harvesting *Spirulina*, as it is a low-cost, energy-efficient, and easily implementable method [38] compared to centrifugation, which is highly effective but energy-intensive [39], filtration, which requires frequent maintenance due to clogging [40], and flocculation-assisted sedimentation, which, while cost-effective, requires additional processing to remove residual flocculants [41]. Many studies of the morphological conversion of different *A. platensis* strains have focused on changes in morphology, ultrastructure, physiology, biochemistry, and genetics [42,43]. However, no research has yet examined the relationship of harvest efficiency with *A. platensis* morphology. In this study, we examined the relationship of filament length of *A. platensis* with different types and concentrations of sodium compounds in the growth medium, and then examined the effect of filament morphology on the harvest efficiency using filtration and auto-flotation. We also compared the linear and coiled types of *A. platensis* in terms of growth rate, photosynthetic activity, and auto-flotation efficiency. Our results provide a foundation that led to a hypothesis that explains the relative rarity of linear filaments in nature.

## 2. Materials and Methods

### 2.1. Microorganism and Culture Conditions

*Arthrospira platensis* KCTC AG40101 was cultivated under different conditions, and the effect of the type and concentration of the carbonate source on filament morphology was determined. A spontaneously converted linear *A. platensis* isolate was screened from the coiled wild-type *A. platensis*. Both strains were grown in the original SOT medium to compare their growth rate, harvesting efficiency, and photosynthetic performance: 16.8 g/L NaHCO_3_, 0.5 g/L K_2_HPO_4_, 2.5 g/L NaNO_3_, 1.0 g/L K_2_SO_4_, 1.0 g/L NaCl, 0.2 g/L MgSO_4_·7H_2_O, 0.04 g/L CaCl_2_·2H_2_O, 0.01 g/L FeSO_4_·7H_2_O, 0.07 g/L Na_2_·EDTA·2H_2_O, and 1 mL of a micronutrient solution. The micronutrient solution consisted of 2.86 g/L H_3_BO_3_, 1.81 g/L MnCl_2_·4H_2_O, 0.222 g/L ZnSO_4_·4H_2_O, 0.021 g/L Na_2_MoO_4_·2H_2_O, 0.08 g/L CuSO_4_·5H_2_O, and 0.05 g/L Co(NO_3_)_2_·6H_2_O. The cells were cultivated in a closed 2 L bottle-type photobioreactor system that had a hydrophobic air filter embedded in the bottom of the reactor. During cultivation, the bioreactor was efficiently mixed with 3% CO_2_ and the photon flux density from a cool white LED lamp was 100 μmol photons/m^2^∙s.

The different types and concentrations of carbonate compounds were 0.1 M NaHCO_3_, 0.4 M NaHCO_3_, and 0.2 M Na_2_CO_3_ (instead of 0.2 M NaHCO_3_ in the original SOT medium). To investigate the effect of the concentration of the sodium ion on filament morphology, 0.2 M NaCl was added to the original SOT medium. A 0.1 L sample of *A. platensis* filaments grown in SOT medium containing 0.2 M NaHCO_3_ was used as a preculture for each cultivation experiment. Subsequently, we observed changes in filament morphology over time under different conditions.

### 2.2. Screening of Linear Filaments

We obtained linear filaments of *A. platensis* by inducing the spontaneous conversion of coiled filaments. For screening of linear filaments, we passed a culture consisting of linear and coiled filaments through a sieve with 30 µm pores. Because the helix diameter of the spiral filaments is greater than 30 µm, most of the coiled filaments remained on top of the sieve. Following several passages through the sieve, filaments that passed through were added into a 96-well microplate. Then, based on microscopic observations, we selected samples that only contained linear filaments for cultivation. We also collected coiled filaments from filaments that remained on top of the sieve. To determine whether the selected linear filaments could revert to their original coiled morphology, all filament samples were observed under a microscope (OLYMPUS CKX41, Tokyo, Japan) during subculturing for one year.

### 2.3. Filament Morphology

The morphological variations in *A. platensis* filaments related with filament length were observed using a hemocytometer (Marienfeld Superior counting chamber) with a 0.1 mm scale. Observations were conducted every three days using an optical microscope (OLYMPUS CKX41, Japan). The lengths of ten randomly selected filaments were measured, and averages and standard deviations were presented.

### 2.4. Monitoring of Cell Growth

For the measuring dry cell weight, the microalgal suspensions were filtered through pre-dried and pre-weighed 0.45 µm cellulose nitrate membranes (Whatman, Cleves, OH, USA), and the cells were dried in an oven at 80 °C for 24 h before the dry cell weight was measured.

### 2.5. Efficiency of Auto-Flotation and Filtration

To measure the auto-flotation efficiency, filaments in the stationary phase were collected and placed into 50 mL self-standing Falcon tubes. The tubes were briefly shaken and then allowed to stand. A 1 mL sample was then taken from a depth of 3 cm after 0, 3, 6, 9, and up to 24 h. To determine the auto-flotation efficiency, a spectrophotometer (JASCO V-730, Tokyo, Japan) was used to measure OD_680nm_ of this sample. The auto-flotation efficiency at each time was then expressed as follows:[OD_680_ (t_0_) − OD_680_ (t_i_)]/[OD_680_ (t_0_)] × 100%,(1)
where OD_680_ (t_0_) is the initial cell concentration and OD_680_ (t_i_) is the cell concentration at time t_i_.

Filtration efficiency was determined by passing coiled and linear filaments through standard sieves with pore sizes of 0.1 and 0.075 mm. The cell concentration (OD_680 nm_) was measured using a spectrometer (JASCO V-730), and filtration efficiency was expressed as follows:[OD_680_ (t_0_) − OD_680_ (t_f_)]/[OD_680_ (t_0_)] × 100%,(2)
where OD_680_ (t_0_) is the initial cell concentration (before filtration) and OD_680_ (t_f_) is the cell concentration after filtration.

### 2.6. Biomass and Photosynthetic Activity

The in vivo photosynthetic activity of filaments was determined by measurement of chlorophyll fluorescence with a Multi-Color-PAM fluorimeter (Heinz Walz, Effeltrich, Germany) after dark adaptation of filaments for 20 min. A stepwise increase in the level of the actinic LED light (440 nm), with a step width of 1 min, was used to record light response curves. The relative electron transport rate (rETR) and effective quantum yield of PSII (Y(II)) were calculated.

## 3. Results and Discussion

### 3.1. Effect of NaHCO_3_ Concentration on Growth and Morphology

We first investigated the effect of NaHCO_3_ concentration on the growth and morphology of *A. platensis* by culturing filaments in the standard SOT medium (with 0.2 M NaHCO_3_), in medium with 0.1 M NaHCO_3_, and in medium with 0.4 M NaHCO_3_ (Figure 1).

The results show no significant differences in cell growth when the medium had 0.1 M and 0.2 M NaHCO_3_, but the growth rate was slightly delayed when the medium had 0.4 M NaHCO_3_, although the biomass on day 21 was similar in all three groups (Figure 1a). The cultures using 0.1 M NaHCO_3_, 0.2 M NaHCO_3_, and 0.4 M NaHCO_3_ maintained relatively constant pH values of 8.46, 8.66, and 8.91 on average during cultivation (Figure 1b). The NaHCO_3_ concentration also affected filament length. In particular, the average filament length was similar when the medium had 0.1 M or 0.2 M NaHCO_3_, and there were no significant changes during the 21-day culture period. However, when the medium had 0.4 M NaHCO_3_, the filament length increased significantly on day 3 (from 0.2950 mm to 0.4180 mm) and the length reached 0.6498 mm on day 6, more than twice the initial length (Figure 1c). The filament length continued to increase gradually until day 15 and then decreased slightly on day 18 and day 21 (Figure 1c). This decrease in filament length after day 18 is likely attributable to reduction of NaHCO_3_ in the growth medium as a result of cellular growth. These results show that the NaHCO_3_ concentration affected the morphology of *A. platensis* filaments, in that increasing concentration of NaHCO_3_ could increase the filament length of *A. platensis* (Figure 1c).

### 3.2. Effect of Sodium Ion Concentration on Growth and Morphology

We then examined whether the increased length of *A. platensis* filaments, when grown with 0.4 M NaHCO_3_, was due to the increased sodium concentration. Thus, we initially cultured filaments in the standard SOT medium (which contains 0.2 M NaHCO_3_), transferred them into medium with 0.2 M NaCl + 0.2 M NaHCO_3_, and then determined changes in growth and morphology.

The results show that filament length was greater beginning on day 6 when the filaments were grown in medium that contained 0.2 M NaCl + 0.2 M NaHCO_3_, reaching a maximum of approximately 0.7161 mm on day 12, but decreasing slightly thereafter (Figure 2c). This response is similar to the effect of 0.4 M NaHCO_3_ (Figure 1c) and indicates that increased concentration of the sodium ion was responsible for the elongation of *A. platensis* filaments.

However, we also observed distinct differences in filament morphology when filaments were grown with a 0.2 M NaCl + 0.2 M NaHCO_3_ rather than 0.4 M NaHCO_3_. In particular, in the medium with 0.2 M NaCl, the filament length increased as the filament width decreased. Previous studies reported similar effects of sodium concentration on the spiral structure of *A. platensis* filaments. For example, low salinity conditions (13 g/L) in the presence of different sodium salts (NaHCO_3_, NaCl, Na_2_SO_4_) led to the observation of abnormally long trichomes due to salinity stress, whereas high-salinity NaCl-based media (55–68 g/L) resulted in short and dense spiral structures [24]. Additionally, another study demonstrated that helix length, diameter, and pitch length were affected by the light spectrum, NaCl concentration, and glucose level [25]. In the present study, sodium-mediated effects on filament length affected the harvest efficiency of *A. platensis* by altering auto-flotation, as described in more detail below.

We also examined the effect of sodium ion concentration on the growth of *A. platensis*. Throughout the cultivation period, the addition of 0.2 M NaCl did not cause noticeable pH changes compared to the control condition (Figure 2b). But the use of high concentrations of NaCl significantly inhibited the growth beginning on day 12 (Figure 2a). This negative effect may be attributed to the reduced efficiency of light utilization caused by salt stress, specifically the presence of Cl^−^ ions [24]. Thus, these experiments confirmed that increasing concentration of NaCl could increase the filament length of *A. platensis* (Figure 2c) but also inhibited cell growth, making this medium unsuitable for culturing of *A. platensis*.

### 3.3. Effect of Cultivation of Using 0.2 M Na_2_CO_3_ as a Carbonate Source

As the length of *A. platensis* filaments increase, the efficiency of filtration harvesting also increases. When 0.2 M NaCl was added to the standard SOT medium, the length of filaments increased, but growth was greatly inhibited and the filaments clumped together, making sustainable culture problematic (Figure 2). Similarly, although cultivation in a medium with 0.4 M NaHCO_3_ led to increased filament length (Figure 1), it required a doubling of the carbonate source, making cultivation less cost-effective. To address this issue, we measured the growth and morphology of filaments cultured in medium in which the standard 0.2 M NaHCO_3_ was replaced with 0.2 M Na_2_CO_3_ (an equivalent molar concentration of sodium ions as medium with 0.4 M NaHCO_3_).

Our results show that filament length was similar when *A. platensis* was grown in medium with 0.2 M Na_2_CO_3_ and medium with 0.4 M NaHCO_3_ or 0.2 M NaCl + 0.2 M NaHCO_3_ (Figure 1c, Figure 2c and Figure 3c). In particular, filament length (0.4634 mm) more than doubled by day 3 compared to day 0, continued to increase, and reached a maximum of 0.7200 mm on day 9 (Figure 3c). In addition, the growth rate in the medium with 0.2 M Na_2_CO_3_ was comparable to the growth rate in the medium with 0.4 M NaHCO_3_ (Figure 1a and Figure 3a). The initial pH of the culture condition using 0.2 M Na_2_CO_3_ was high at 10.02. However, due to aeration with 3% carbon dioxide, the average pH during the cultivation period was 8.92, which was not significantly higher than that of the control medium condition using 0.2 M NaHCO_3_ (Figure 3b). Therefore, replacing 0.2 M NaHCO_3_ with 0.2 M Na_2_CO_3_ in the standard SOT medium led to a more than a two-fold increase in filament length and did not significantly inhibit growth. These findings suggest that the concentration of sodium ions is directly related to the length of *A. platensis* filaments.

Conclusively, the results examining the morphological changes in *Spirulina* under different NaHCO_3_ concentrations showed that while pH values increased sequentially with higher NaHCO_3_ concentrations, the differences were not significant (Figure 1b). Additionally, in conditions where 0.2 M NaCl was added and 0.2 M Na_2_CO_3_ was used, no significant pH variation was observed compared to the control condition (Figure 2b and Figure 3b). Through this, we can conclude that under the experimental conditions used in this study, pH changes did not affect the variation in *Spirulina*’s filament length. The increase in sodium ions resulting from the use of NaHCO_3_ concentration in the medium is accompanied by a rise in HCO_3_^−^ concentration. Therefore, it is difficult to attribute the elongation of *Spirulina* filaments solely to the effect of Na^+^ ions. However, the elongation of *Spirulina* filaments observed in both the condition where only 0.2 M NaCl was added to the control medium and the condition where 0.2 M Na_2_CO_3_ was used as a carbonate source suggests that the increase in Na^+^ concentration is a key factor influencing *Spirulina*’s morphological changes. Many cyanobacteria, including *Spirulina*, are known to undergo morphological changes due to osmotic stress under high Na^+^ concentration conditions [24,44,45,46]. In the culture conditions using 0.4 M NaHCO_3_, 0.2 M Na_2_CO_3_, and 0.2 M NaCl, a slight decrease in *Spirulina* biomass accumulation was observed after the initial cultivation phase compared to the control condition. However, since the filament length of *Spirulina* increased under these conditions, it is likely that the elongation resulted from salt stress caused by the increased Na^+^ ion concentration, which inhibited cell division.

### 3.4. Effect of Different Carbonates on Harvest Efficiency

The harvesting of microalgae accounts for 20 to 30% of the total cost of biomass production when using energy-intensive methods, such as centrifugation [47,48]. Therefore, optimizing harvesting methods based on filtration and flotation is critically important for decreasing energy consumption and harvest time and for enabling recycling of growth media. We therefore evaluated the impact of variations in morphology and filament length in *A. platensis*, which were induced by adjusting the sodium cation, on the efficiency of harvest via filtration and flotation.

Thus, we examined the effect of mesh sizes of 0.100 mm and 0.075 mm to consider the efficiency of water drainage, a variable related to harvest time (Figure 4).

For a medium that increased filament length to 0.7 mm or more (0.4 M NaHCO_3_, 0.2 M NaCl + 0.2 M NaHCO_3_, 0.2 M Na_2_CO_3_), the harvesting efficiency was more than 93% for both mesh sizes. As expected, the harvesting efficiencies for filaments cultured in medium with 0.1 or 0.2 M NaHCO_3_ conditions were much lower (57% and 78% when using the 0.100 mm mesh). These results demonstrate that a harvest efficiency of more than 90% with a 0.100 mm mesh can be achieved when the filament length is at least 0.7 mm.

We also evaluated harvest efficiency using a flotation method, in which 50 mL of filaments grown under different conditions were transferred into a self-standing Falcon tube, thoroughly mixed, and then allowed to stand so the filaments could float. We then measured OD_680nm_ of 1 mL samples collected from the midpoint of the Falcon tube and calculated flotation efficiency as the ratio of filament concentration at time *t* to the initial filament concentration. The results show that harvest efficiency by flotation increased with as the total sodium ion concentration increased (Figure 5a).

Moreover, when the medium had 0.4 M NaHCO_3_ or 0.2 M Na_2_CO_3_ (conditions that led to the longest filaments), harvest efficiency exceeded 90% within 2 h. The lowest flotation efficiency occurred in medium with 0.1 M NaHCO_3_ (below 60% even after 2 h). However, the salinity level can also increase the buoyancy of an object. Therefore, it cannot be definitively concluded that changes in *A. platensis* filament length, which determined sodium concentration, was the sole factor that affected harvest efficiency.

Considering the increased cost associated with the use of additional medium components, our results demonstrate that a growth medium with 0.2 M Na_2_CO_3_ was most effective for maximizing the efficiency of auto-flotation. Additionally, our experiments with the 0.2 M NaCl + 0.2 M NaHCO_3_ medium suggest that although this medium led to increased filament length, it also decreased flotation efficiency, presumably due to internal morphological changes caused by salt stress.

### 3.5. Photosynthesis and Biomass Accumulation of Linear Filaments

Notably, our microscopic observations of samples of linear filaments and coiled filaments (Figure 6a,b) showed that when these samples had the same optical density (OD_680nm_ = 0.5), there were about two-fold more linear filaments than coiled filaments (Figure 6).

To compare light penetration of cultures of linear and coiled filaments, we set up two transparent rectangular bottles (50 × 100 × 30 cm) in front of a light panel, so that the photon flux density was 100 µmol photons m^−2^ s^−1^ at the outer surface of each bottle. Then, we adjusted the concentration of cells in each bottle so that the transmitted light had a photon flux density of 10 µmol photons m^−2^ s^−1^. Under these conditions, the bottle with linear filaments had 104 ± 5 filaments/mL and the bottle with coiled filaments had 62 ± 5 filaments/mL. In other words, the linear filaments allowed more light penetration, decreased self-shading, and had less light scattering due to their narrow linear shape. Culture density and light absorbance are related to the yield during photoautotrophic culturing in regard to the total volume of the photobioreactor. The cellular growth of the two filament types was compared under the same growth conditions. Other research reported that linear filaments of *A. platensis* had a lower growth yield than spiral filaments [23]. However, our results showed that the linear filaments had a faster growth rate than the coiled filaments (Figure 7).

This result is similar to the results from the analysis of a *Synechocystis* Olive mutant that had a partially truncated light harvesting antenna, which increased penetration depth of light into the culture volume and consequently increased time–space productivity [49].

Moreover, our linear filaments had increased photosynthetic activity based on measurements of the relative electron transport rate and quantum yield of PSII (Figure 8). This implies that the linear filaments had more resistance to high light levels than the coiled filaments. It is interesting that the coiled and linear filaments had no significant difference in the maximum quantum yield of PSII at very low photon flux density, which implies similar PSII status of these different filaments at low light levels. Thus, the higher photosynthetic efficiency of linear filaments is likely due to the greater light penetration throughout the culture.

The morphological conversion of coiled filaments to linear filaments was mainly believed to be heritable and stable due to genetic drift [23,50]. No studies have reported linear *A. platensis* filaments in nature, although linearization of spiral filaments frequently occurs in laboratories and mass cultures. Wang and Zhao reported that linear filaments of *A. platensis* could revert to the original coiled morphology under certain conditions [23]. However, spontaneous conversion of linear filaments to coiled filaments based on genetic changes cannot explain the dominance of the coiled form in nature due to the long process of *A. platensis* evolution. Moreover, in our experiment, following this screening procedure, we observed no morphological reversion from the linear to coiled form, although linear filaments were readily generated from coiled filaments. In the present study, we found distinct differences in auto-flotation activity of linear and coiled filaments, and it is likely that the greater buoyancy of coiled filaments gives them survival advantages in nature.

### 3.6. Auto-Flotation Efficiency of Linear and Coiled Filaments

It is reasonable to speculate the linearization of *A. platensis* could also occur in nature. Natural aquatic environments differ from the culture conditions in a photobioreactor, in which artificial mixing is achieved by shaking or airlifting. *A. platensis* filaments are inherently buoyant due to their gas vesicles [51]. In competition among photosynthetic microorganisms in nature, migration to the water surface will lead to increased photosynthesis and growth. The flotation activity of *A. platensis* could be affected by physiological parameters.

We also investigated the effect of filament morphology on auto-flotation over a period of 24 h (Figure 9). The coiled filaments achieved an auto-floating efficiency of about 90%. In contrast, the linear filaments achieved a maximum auto-floating efficiency of only about 10%, even after 24 h, presumably because these filaments are thin (diameter of 6–12 μm [52]) and have a compact structure. In fact, when linear filaments and coiled filaments are mixed and shaken, the linear filaments mostly accumulate in the middle layer, while the coiled filaments accumulate in the upper layer.

Consequently, linear filaments of *A. platensis* could be excluded by competition with coiled filaments soon after their formation due to their poor auto-flotation in spite of their better light utilization due to less auto-shading. However, this assumption is based on results obtained in a controlled environment and may not fully reflect *Spirulina*’s behavior in more complex natural conditions. Due to its distinct morphology, linear *Spirulina* achieves higher photosynthetic efficiency, enabling greater biomass accumulation. However, it exhibits significantly lower harvesting efficiency through filtration or auto-flotation. In the commercialization of linear or coiled *Spirulina*, the benefits of enhanced harvesting efficiency in the coiled form significantly outweigh the biomass accumulation losses compared to the linear type.

## 4. Conclusions

This study demonstrated that an increase in the concentration of sodium ions in the growth medium led to increases in the length of linear filaments of *A. platensis* and associated improvements in harvest efficiency by filtration and auto-flotation. Specifically, an increase in the sodium ion concentration from 0.2 M to 0.4 M led to a more than doubling of filament length (from 0.3393 mm to 0.7084 mm), and this increase in filament length led to increased filtration efficiency through a standard 0.100 mm sieve (from 78.53% to 93.85%). Linear filaments of *A. platensis* had superior growth and photosynthetic efficiency compared to coiled filaments in our laboratory experiments. However, linear filaments also have lower buoyancy, so the coiled filaments tend to dominate at the water surface in natural environments, where the light level is greatest. This may explain the dominance of coiled filaments of *A. platensis* in natural ecosystems.

## Figures and Tables

**Figure 1 microorganisms-13-00367-f001:**
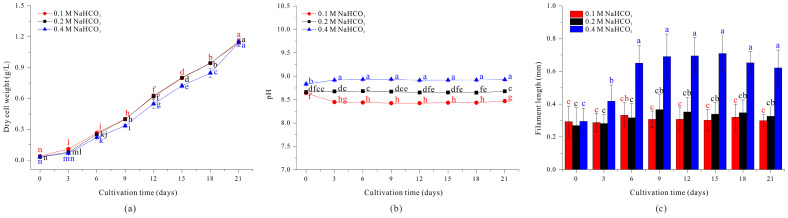
Effect of NaHCO_3_ concentration on the growth (**a**), pH (**b**), and length (**c**) of *A. platensis* filaments. The data were analyzed using one-way ANOVA, followed by Duncan’s test (*p* < 0.05). Lowercase letters indicate statistically significant differences among groups.

**Figure 2 microorganisms-13-00367-f002:**
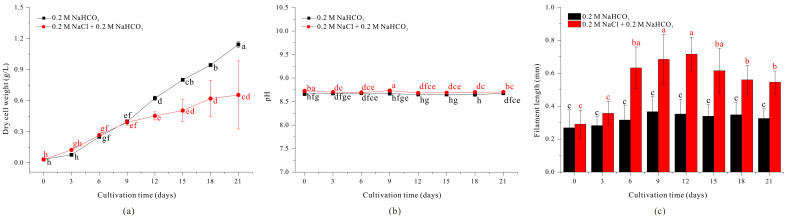
Effect of 0.2 M NaHCO_3_ (black) and 0.2 M NaCl + 0.2 M NaHCO_3_ (red) on growth (**a**), pH (**b**), and length (**c**) of *A. platensis* filaments. The data were analyzed using one-way ANOVA, followed by Duncan’s test (*p* < 0.05). Lowercase letters indicate statistically significant differences among groups.

**Figure 3 microorganisms-13-00367-f003:**
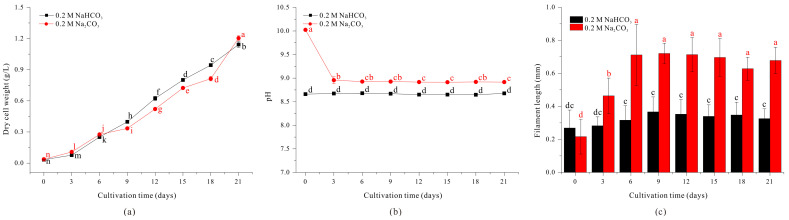
Effect of 0.2 M NaHCO_3_ (black) and 0.2 M Na_2_CO_3_ (red) on the growth (**a**), pH (**b**) and the length (**c**) of *A. platensis* filaments. The data were analyzed using one-way ANOVA, followed by Duncan’s test (*p* < 0.05). Lowercase letters indicate statistically significant differences among groups.

**Figure 4 microorganisms-13-00367-f004:**
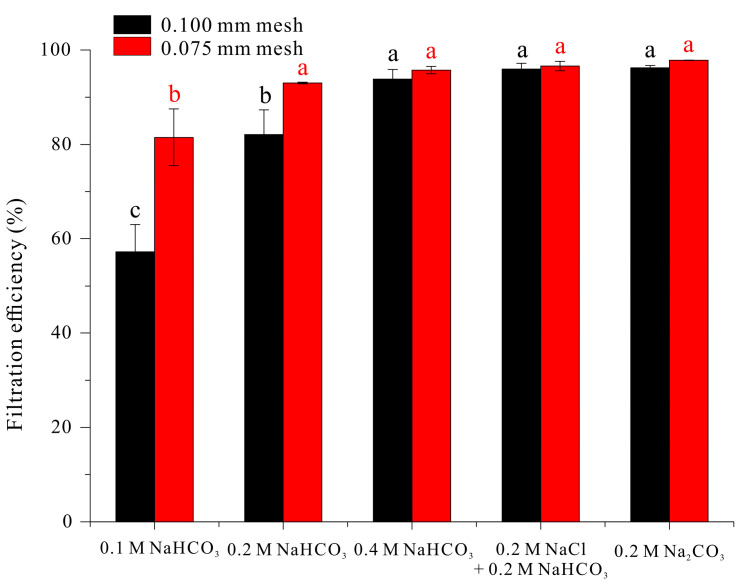
Effect of different carbonates on filtration efficiency of *A. platensis* filaments. The data were analyzed using one-way ANOVA, followed by Duncan’s test (*p* < 0.05). Different lowercase letters at the top of the bar graph indicate statistically different between groups (*p* < 0.05), and the same lowercase letter indicates no significant difference (*p* > 0.05). n = 3.

**Figure 5 microorganisms-13-00367-f005:**
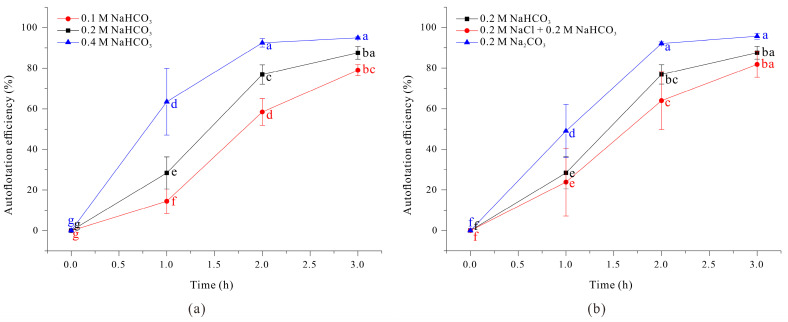
Effect of NaHCO_3_ concentration (**a**) and different sodium compounds (**b**) on auto-floatation efficiency of *A. platensis* filaments. The data were analyzed using one-way ANOVA, followed by Duncan’s test (*p* < 0.05). Lowercase letters indicate statistically significant differences among groups.

**Figure 6 microorganisms-13-00367-f006:**
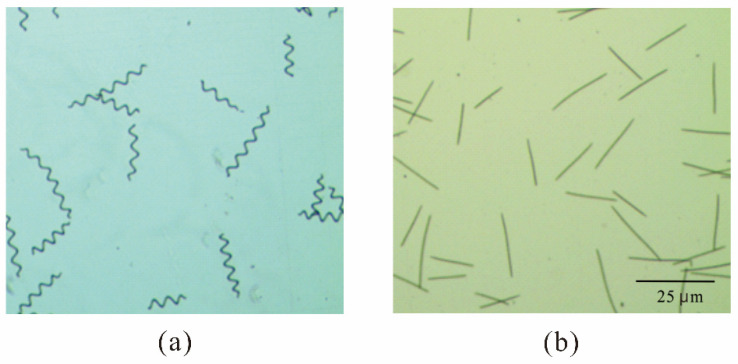
Microscopy of coiled filaments (**a**) and linear filaments (**b**) of *A. platensis* cultures, each with an absorbance of 0.5 at 680 nm.

**Figure 7 microorganisms-13-00367-f007:**
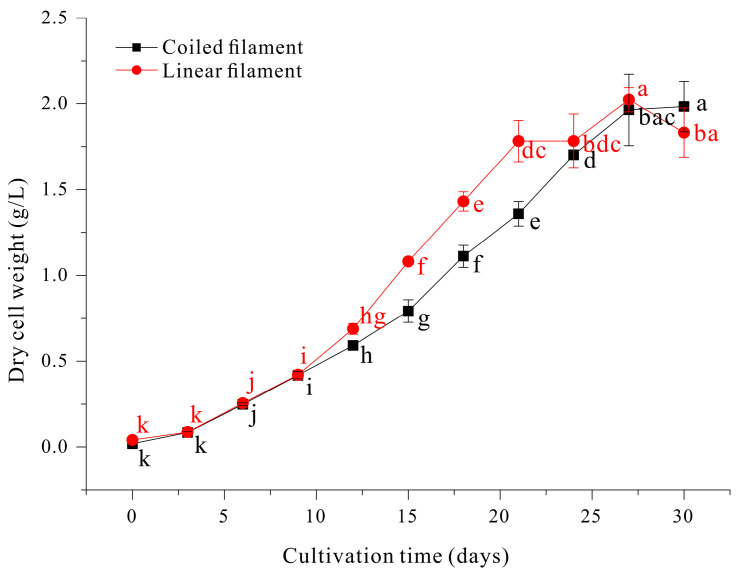
Growth of coiled filaments and linear filaments of *A. platensis*. The data were analyzed using one-way, ANOVA followed by Duncan’s test (*p* < 0.05). Lowercase letters indicate statistically significant differences among groups.

**Figure 8 microorganisms-13-00367-f008:**
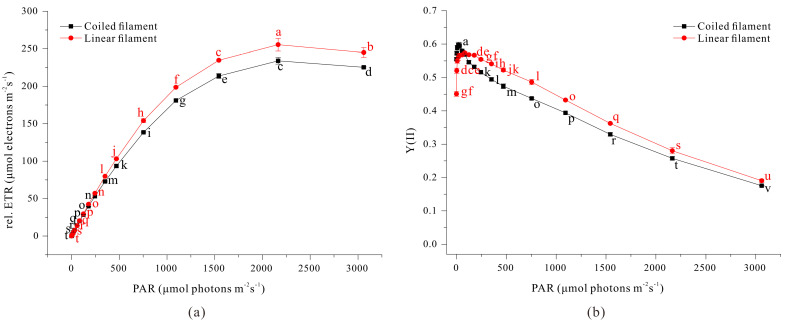
Effect of the photon flux density of photosynthetically active radiation on the relative electron transport rate (**a**) and the effective quantum yield of PSII (**b**) in coiled and linear filaments of *A. platensis*. The data were analyzed using one-way ANOVA, followed by Duncan’s test (*p* < 0.05). Lowercase letters indicate statistically significant differences among groups.

**Figure 9 microorganisms-13-00367-f009:**
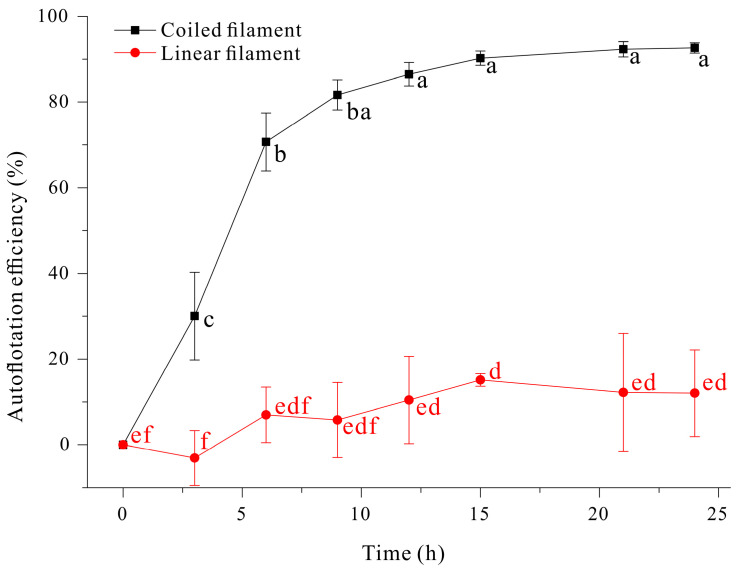
Effect of incubation time on auto-floatation efficiency of coiled and linear filaments of *A. platensis*. The data were analyzed using one-way ANOVA, followed by Duncan’s test (*p* < 0.05). Lowercase letters indicate statistically significant differences among groups.

## Data Availability

The original contributions presented in this study are included in the article. Further inquiries can be directed to the corresponding author.

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
