# Peer review of "Relationship Between Harvesting Efficiency and Filament Morphology in Arthrospira platensis Gomont"

_microorganisms, 2025, doi:10.3390/microorganisms13020367_

Round 1
Reviewer 1 Report
Comments and Suggestions for Authors
This manuscript entitled “Relationship between harvesting efficiency and filament morphology in Arthrospira platensis” presents a detailed examination of the influence of sodium ion concentration on the filament length of A. platensis and harvesting efficiency of this well-studied cyanobacterium. Besides that, the growth, photosynthetic activity, and autoflotation efficiency of coiled and linear filaments of A. platensis were studied.
However, here are some major problems:
1. It is necessary to exclude the effects of HCO3-, CO32-, and Cl-, as well as the pH changes followed by the addition of NaHCO3 and Na2CO3, on the growth and filement lengths of A. platensis. After that it is more reasonable to say the “an increase in the concentration of sodium ions in the led to increases in the length of linear filaments of A. platensis and associated increases in harvest efficiency”.
2. It is necessary to give an explanation for why an increase of sodium ions (or HCO3-/CO32-) can lead to an elongation of filaments and the harvarst efficiency.
3. It is recommended to give an introduction of current harvest methods for A. platensis in both laboratories and industries. Besides, it is better to compare the efficiency, feasibility, and cost of auto-flotation associated harvesting method in this study with other harvesting methods.
4. It is recommended to remove section 3.5 to the Materials and Methods part, thus giving a detailed description of how the linear filaments were selected.
5. For section 2.3, how many replicates were included when measuring OD of filamentous A. platensis? A dry weight curve should be more accurate to reflect the growth condition.
6. In Fig. 1b, it is shown that on Day 21, the filaments of A. platensis under 0.4 M NaHCO3 were much high than that under 0.1 M or 0.2 M NaHCO3. At the same time, OD680 of cultures under the corresponding treatment were the same (Fig. 1a). Here may need further explanation.
7. For Fig. 3, it is necessary to compare the difference of HCO3- and CO32- because they have a different capacity to neutralize H+. It is better to also present the pH change during cultivation.
8. Why not test a series of NaCl concentration instead of only choosing 0.2 M NaCl (Fig. 2,5)?
9. All the figures in this manuscript need statistical analysis, as well as greater resolution.
10. It is recommended to cite more publications in the last five years as references.
Comments on the Quality of English LanguageIt is recommended to improve the English expression in the manuscript to give a clearer and sounder presentation of this study.
Author Response
Reviewer 1
This manuscript entitled “Relationship between harvesting efficiency and filament morphology in Arthrospira platensis” presents a detailed examination of the influence of sodium ion concentration on the filament length of A. platensis and harvesting efficiency of this well-studied cyanobacterium. Besides that, the growth, photosynthetic activity, and autoflotation efficiency of coiled and linear filaments of A. platensis were studied.
However, here are some major problems:
- It is necessary to exclude the effects of HCO3-, CO32-, and Cl-, as well as the pH changes followed by the addition of NaHCO3 and Na2CO3, on the growth and filament lengths of A. platensis. After that it is more reasonable to say the“an increase in the concentration of sodium ions in the led to increases in the length of linear filaments of A. platensis and associated increases in harvest efficiency”.
Sol) We appreciate your comment. We added a graph showing pH changes under different culture conditions (Fig. 1, 2, and 3). We revised the corresponding figures and added the sentence about pH value.
The cultures using 0.1 M NaHCO₃, 0.2 M NaHCO₃, and 0.4 M NaHCO₃ maintained relatively constant pH values of 8.46, 8.66, and 8.91 on average during cultivation (Fig. 1B)
The revised parts are highlighted in red color in the text of this manuscript (Line 153 - 154 on Page 4).
Throughout the cultivation period, the addition of 0.2 M NaCl did not cause noticeable pH changes (pH 8.66) compared to the control condition (Fig. 2b).
The revised parts are highlighted in red color in the text of this manuscript (Line 194 - 195 on Page 5).
The initial pH of the culture condition using 0.2 M Na₂CO₃ was high at 10.02. However, due to aeration with 3% carbon dioxide, the average pH during the cultivation period was 8.92, which was not significantly higher than that of the control medium condition using 0.2 M NaHCO₃ (Fig. 3B).
The revised parts are highlighted in red color in the text of this manuscript (Line 216 - 220 on Page 5 - 6).
As the reviewer pointed out, it is important to explain the effects of pH on changes in Spirulina morphology.
We added this following discussion to the main text.
“Conclusively, the results examining the morphological changes of Spirulina under different NaHCO₃ concentrations showed that while pH values increased sequentially with higher NaHCO₃ concentrations, the differences were not significant (Fig. 1b). Additionally, in conditions where NaCl was added and Na₂CO₃ was used, no significant pH variation was observed compared to the control condition (Fig. 2b and 3b). Through this, we can conclude that under the experimental conditions used in this study, pH changes did not affect the variation in Spirulina's filament length.”
The revised parts are highlighted in red color in the text of this manuscript (Line 229 - 235 on Page 6).
As the reviewer pointed out, it is also important to explain the effects of HCO₃⁻, CO₃²⁻, and Cl⁻ on changes in Spirulina morphology.
“Conclusively, the results examining the morphological changes of Spirulina under different NaHCO₃ concentrations showed that while pH values increased sequentially with higher NaHCO₃ concentrations, the differences were not significant (Fig. 1b). Additionally, in conditions where 0.2 M NaCl was added and 0.2 M Na₂CO₃ was used, no significant pH variation was observed compared to the control condition (Fig. 2b and 3b). Through this, we can conclude that under the experimental conditions used in this study, pH changes did not affect the variation in Spirulina's filament length. The increase in sodium ions resulting from the use of NaHCO₃ concentration in the medium is accompanied by a rise in HCO₃⁻ concentration. Therefore, it is difficult to attribute the elongation of Spirulina filaments solely to the effect of Na⁺ ions. However, the elongation of Spirulina filaments observed in both the condition where only 0.2 M NaCl was added to the control medium and the condition where 0.2 M Na₂CO₃ was used as a carbonate source suggests that the increase in Na⁺ concentration is a key factor influencing Spirulina's morphological changes.”
We added this discussion to the main text.
The revised parts are highlighted in red color in the text of this manuscript (Line 229 - 242 on Page 6).
- It is necessary to give an explanation for why an increase of sodium ions (or HCO3-/CO32-) can lead to an elongation of filaments and the harvarst efficiency.
Sol) Under the experimental conditions used in this study, we can conclude that the increase in Na⁺ concentration is a key factor influencing Spirulina's morphological changes.
“Many cyanobacteria, including Spirulina, are known to undergo morphological changes due to osmotic stress under high Na⁺ concentration conditions [24,44-46]. In the culture conditions using 0.4 M NaHCO₃, 0.2 M Na₂CO₃, and 0.2 M NaCl, a slight decrease in Spir-ulina biomass accumulation was observed after the initial cultivation phase compared to the control condition. However, since the filament length of Spirulina increased under these conditions, it is likely that the elongation resulted from salt stress caused by the in-creased Na⁺ ion concentration, which inhibited cell division.”
We added this discussion to the main text.
The revised parts are highlighted in red color in the text of this manuscript (Line 242 - 248 on Page 6).
- It is recommended to give an introduction of current harvest methods for A. platensis in bothlaboratories and industries. Besides, it is better to compare the efficiency, feasibility, and cost of auto-flotation associated harvesting method in this study with other harvesting methods.
Sol) Current harvesting methods for Spirulina (A. platensis) vary based on the scale of pro-duction. For laboratory-scale applications, common methods include filtration using membrane filters or mesh sieves [26,27], low-speed centrifugation [28], and sedimentation with or without flocculants [29,30]. In industrial-scale operations, more advanced tech-niques such as microfiltration [31], industrial centrifugation [32], flocculation-assisted sedimentation [33], belt and rotary drum filtration [34,35], and electrocoagulation [36,37] are employed to efficiently separate biomass while ensuring cost-effectiveness. The choice of harvesting method depends on factors such as scalability, energy consumption, and processing efficiency. Among these methods, auto-flotation offers significant advantages for harvesting Spirulina, as it is a low-cost, energy-efficient, and easily implementable method [38] compared to centrifugation, which is highly effective but energy-intensive [39]; filtration, which requires frequent maintenance due to clogging [40]; and floccula-tion-assisted sedimentation, which, while cost-effective, requires additional processing to remove residual flocculants [41]
We added this explanation of current harvest methods for A. platensis to the introduction part of this manuscript.
The revised parts are highlighted in red color in the text of this manuscript (Line 51 - 64 on Page 2).
[Reference]
[26] Ismail, I.; Kurnia, K.A.; Samsuri, S.; Bilad, M.R.; Marbelia, L.; Ismail, N.M.; Khan, A.L.; Budiman, A.; Susilawati, S. Energy efficient harvesting of Spirulina sp. from the growth medium using a tilted panel membrane filtration. Bioresource Technology Reports 2021, 15, 100697.
[27] Cuellar-Bermudez, S.P.; Kilimtzidi, E.; Devaere, J.; Goiris, K.; Gonzalez-Fernandez, C.; Wattiez, R.; Muylaert, K. Harvesting of Arthrospira platensis with helicoidal and straight trichomes using filtration and centrifugation. Separation Science and Technology 2020, 55, 2381-2390.
[28] Yong, T.C.; Chiu, P.-H.; Chen, C.-H.; Hung, C.-H.; Chen, C.-N.N. Disruption of thin-and thick-wall microalgae using high pressure gases: Effects of gas species, pressure and treatment duration on the extraction of proteins and carotenoids. Journal of bioscience and bioengineering 2020, 129, 502-507.
[29] Chatsungnoen, T.; Chisti, Y. Harvesting microalgae by flocculation–sedimentation. Algal Research 2016, 13, 271-283.
[30] Leite, L.d.S.; Daniel, L.A. Optimization of microalgae harvesting by sedimentation induced by high pH. Water Science and Technology 2020, 82, 1227-1236.
[31] Melanie, H.; Aspiyanto, A.; Maryati, Y.; Susilowati, A.; Mulyani, H.; Filailla, E.; Devi, A.F.; Budiari, S. Recovery of phycocyanin from microalgae Spirulina using membrane technology and evaluation of its antioxidant activity. In Proceedings of the AIP Conference Proceedings, 2023.
[32] Zhu, J.; Wakisaka, M.; Omura, T.; Yang, Z.; Yin, Y.; Fang, W. Advances in industrial harvesting techniques for edible microalgae: Recent insights into sustainable, efficient methods and future directions. Journal of Cleaner Production 2024, 140626.
[33] Liu, Z.; Hao, N.; Hou, Y.; Wang, Q.; Liu, Q.; Yan, S.; Chen, F.; Zhao, L. Technologies for harvesting the microalgae for industrial applications: Current trends and perspectives. Bioresource Technology 2023, 129631.
[34] Silva, N.C.; Machado, M.V.; Brandão, R.J.; Duarte, C.R.; Barrozo, M.A. Dehydration of microalgae Spirulina platensis in a rotary drum with inert bed. Powder Technology 2019, 351, 178-185.
[35] Silva, N.C.; Duarte, C.R.; Barrozo, M.A. Analysis of the use of a non-conventional rotary drum for dehydration of microalga Spirulina platensis. Bioprocess and biosystems engineering 2020, 43, 1359-1367.
[36] de Souza Sossella, F.; Rempel, A.; Nunes, J.M.A.; Biolchi, G.; Migliavaca, R.; Antunes, A.C.F.; Costa, J.A.V.; Hemkemeier, M.; Colla, L.M. Effects of harvesting Spirulina platensis biomass using coagulants and electrocoagulation–flotation on enzymatic hydrolysis. Bioresource Technology 2020, 311, 123526.
[37] Krishnamoorthy, N.; Unpaprom, Y.; Ramaraj, R.; Maniam, G.P.; Govindan, N.; Arunachalam, T.; Paramasivan, B. Recent advances and future prospects of electrochemical processes for microalgae harvesting. Journal of Environmental Chemical Engineering 2021, 9, 105875.
[38] Kim, S.G.; Choi, A.; Ahn, C.Y.; Park, C.S.; Park, Y.H.; Oh, H.M. Harvesting of Spirulina platensis by cellular flotation and growth stage determination. Letters in applied microbiology 2005, 40, 190-194.
[39] Najjar, Y.; Abu-Shamleh, A. Harvesting of microalgae by centrifugation for biodiesel production: a review. Algal Res 51: 102046. 2020.
[40] Mkpuma, V.O.; Moheimani, N.R.; Ennaceri, H. Microalgal dewatering with focus on filtration and antifouling strategies: A review. Algal Research 2022, 61, 102588.
[41] Vermuë, M.; Eppink, M.; Wijffels, R.; Van Den Berg, C. Multi-product microalgae biorefineries: from concept towards reality. Trends in biotechnology 2018, 36, 216-227
- It is recommended to remove section 3.5 to the Materials and Methods part, thus giving a detailed description of how the linear filaments were selected.
Sol) We moved the content of Section 3.5 to Section 2.2 in the Materials and Methods
- For section 2.3, how many replicates were included when measuring OD of filamentous A. platensis? A dry weight curve should be more accurate to reflect the growth condition.
Sol) The experiments about cellular growth were performed in triplicate. All growth-related values of Spirulina have been replaced with dry cell weight (see Fig. 1a, 2a, 3a, and 7 ). Data were analyzed using one-way ANOVA followed by Duncan’s test (p <0.05).
- In Fig. 1b, it is shown that on Day 21, the filaments of A. platensis under 0.4 M NaHCO3 were much high than that under 0.1 M or 0.2 M NaHCO3. At the same time, OD680 of cultures under the corresponding treatment were the same (Fig. 1a). Here may need further explanation.
Sol) I appreciate your comment. This question was greatly helpful in deducing the reason for Spirulina filament elongation caused by the increase in Na⁺ ions. The OD 680 value can be considered proportional to the chlorophyll concentration in Spirulina. The increased filament length of Spirulina under the 0.4 M NaHCO₃ condition, without an increase in Absorbance at 680 nm compared to the control condition, indicates that the number of Spirulina cells did not increase. Therefore, the increase in Spirulina filament length may be due to delayed cell division caused by salt stress resulting from the increased Na⁺ ion concentration.
We added this discussion to the main text.
The revised parts are highlighted in red color in the text of this manuscript (Line 242 - 248 on Page 6).
Please refer to the following sentence.
“Many cyanobacteria, including Spirulina, are known to undergo morphological changes due to osmotic stress under high Na⁺ concentration conditions [24,44-46]. In the culture conditions using 0.4 M NaHCO₃, 0.2 M Na₂CO₃, and 0.2 M NaCl, a slight decrease in Spir-ulina biomass accumulation was observed after the initial cultivation phase compared to the control condition. However, since the filament length of Spirulina increased under these conditions, it is likely that the elongation resulted from salt stress caused by the in-creased Na⁺ ion concentration, which inhibited cell division
- For Fig. 3, it is necessary to compare the difference of HCO3-and CO32- because they have a different capacity to neutralize H+. It is better to also present the pH change during cultivation.
Sol) We agree your comment. We added a graph showing pH changes in Figure 1, 2, and 3.
- Why not test a series of NaCl concentration instead of only choosing 0.2 M NaCl (Fig. 2,5)?
Sol) We appreciate your comment. In the preliminary experiment, no changes in morphology of Spirulina filaments were observed under the 0.1 M NaCl condition. Moreover, to balance the sodium concentration in the control medium containing only 0.2 M NaHCO₃ with that of 0.4 M NaHCO₃ and 0.2 M Na₂CO₃, 0.2 M NaCl was added.
- All the figures in this manuscript need statistical analysis, as well as greater resolution.
Sol) All data in figures were analyzed using one-way ANOVA followed by Duncan’s test (p <0.05).
Please refer to Fig. 1, 2, 3, 4, 5, and 7.
- It is recommended to cite more publications in the last five years as references.
Sol) We have updated the references by replacing outdated papers with the latest ones where applicable and referenced more studies.
[Newly added Literature]
[4] Usharani, G.; Saranraj, P.; Kanchana, D. Spirulina cultivation: a review. Int J Pharm Biol Arch 2012, 3, 1327-1341.
[6] Madhubalaji, C.K.; Rashmi, V.; Chauhan, V.S.; Shylaja, M.D.; Sarada, R. Improvement of vitamin B12 status with Spirulina supplementation in Wistar rats validated through functional and circulatory markers. Journal of food biochemistry 2019, 43, e13038.
[7] Dey, S.; Rathod, V.K. Ultrasound assisted extraction of β-carotene from Spirulina platensis. Ultrasonics sonochemistry 2013, 20, 271-276.
[21] Hongsthong, A.; Sirijuntarut, M.; Prommeenate, P.; Thammathorn, S.; Bunnag, B.; Cheevadhanarak, S.; Tanticharoen, M. Revealing differentially expressed proteins in two morphological forms of Spirulina platensis by proteomic analysis. Molecular biotechnology 2007, 36, 123-130.
[22] Mishra, T.; Joshi, M.; Singh, S.; Jain, P.; Kaur, R.; Ayub, S.; Kaur, K. Spirulina: The Beneficial Algae. Int. J App. Micro. Sci 2013, 2, 21-35.
[43] Hifney, A.F.; Abdel-Wahab, D.A. Phyco-based synthesis of TiO 2 nanoparticles and their influence on morphology, cyto-ultrastructure and metabolism of Spirulina platensis. Rendiconti Lincei. Scienze Fisiche e Naturali 2019, 30, 185-195.
[52] Lecina, M.; Prat, J.; Paredes, C.J.; Cairó, J.J. Non-disruptive sonication of A. fusiformis (A. platensis) cultures facilitates its harvesting. Algal Research 2015, 7, 1-4.
[Revised latest Literaute]
[12] Vasudevan, S.K.; Seetharam, S.; Dohnalek, M.H.; Cartwright, E.J. Spirulina: A daily support to our immune system. International Journal of Noncommunicable Diseases 2021, 6, S47-S54.
[13] Matufi, F.; Maghsudi, H.; Choopani, A. Spirulina and its role in immune system: a review. J Immunol Res Ther 2020, 5, 204-2011.
[18] Janda-Milczarek, K.; Szymczykowska, K.; Jakubczyk, K.; Kupnicka, P.; Skonieczna-Żydecka, K.; Pilarczyk, B.; Tomza-Marciniak, A.; Ligenza, A.; Stachowska, E.; Dalewski, B. Spirulina supplements as a source of mineral nutrients in the daily diet. Applied Sciences 2023, 13, 1011.
[42] Zapata, D.; Arroyave, C.; Cardona, L.; Aristizábal, A.; Poschenrieder, C.; Llugany, M. Phytohormone production and morphology of Spirulina platensis grown in dairy wastewaters. Algal Research 2021, 59, 102469.

Reviewer 2 Report
Comments and Suggestions for Authors
The manuscript is interesting as a prospective study on A. platensis research, which takes on a linear form under laboratory conditions. Several aspects of morphology in relation to certain cultivation conditions have been addressed, including the sodium content from different sources. It is a starting point.
Observations:
The description of the experiments should be more detailed. Specifically, the linear form was isolated based on the cultivation of the Arthrospira platensis KCTC AG40101 strain on a medium containing... Was the inoculum for the experiments cultivated in the same medium?
The addition of Na₂CO₃ and NaCl requires further justification. Na₂CO₃ is more basic than NaHCO₃, so the cultivation medium's pH should be mentioned. NaCl does not alter the pH but increases the Cl⁻ content.
Results of practical importance, such as auto flotation and filtration, are included. The conclusion corresponds mainly to the work but is more of a summary of the experimental observations. An in-depth analysis of the industrial implications of these results would be welcome.
The experimental results come from controlled environments, which may not fully reflect the behavior of the filaments in more complex natural environments. The conclusion suggests spiral filaments dominate in natural environments due to their higher buoyancy. Still, the paper did not detail the competitive advantages of this morphology in terms of survival and ecological adaptation, as it had a different objective.
The study aimed to compare the growth rate, photosynthetic activity, and auto-flotation efficiency of the linear and spiral types of A. platensis. However, the analyzed article does not provide concrete and explicit information about the exact medium in which the spiral forms of A. platensis were cultivated. Were they subjected to the same sodium concentrations? Although comparisons between linear and spiral forms exist, the methodology for spiral forms is not described in detail, and the specific medium in which they were grown is not directly mentioned.
The spiral and linear forms were compared in terms of growth rate, photosynthetic activity, and buoyancy. This suggests that the spiral forms were partially grown in similar media to provide a comparative basis.
Many questions arise; therefore, a detailed description of the experimental design would be beneficial.
The statement that the difference in buoyancy may explain the rarity of linear filaments in natural environments indirectly implies that the morphology of A. platensis filaments (linear or coiled) is reversible or influenced by environmental conditions. However, the article does not explicitly mention the reversibility of linear and coiled forms.
The linear form was obtained by spontaneously converting coiled filaments under laboratory conditions. However, the paper does not clarify whether this change is permanent or if linear filaments can revert to the coiled form under natural conditions.
The differences between the two forms are discussed but without details regarding their reversibility or how these forms coexist or transform in nature.
It is suggested that linear filaments' lower buoyancy makes them less competitive in natural environments, but no experimental data is provided to demonstrate whether linear filaments obtained in the laboratory can revert to the coiled form or vice versa.
From this arises the question: Is the A. platensis KCTC AG40101 strain maintained on the described mineral medium?
Regarding the figures, statistical analysis is not described (information about the number of independent experiments and data processing is missing from the Methods section). The abbreviations (Figures 7 and 8) are clear in that they refer to the form of A. platensis, but this should be explicitly mentioned.
Author Response
Reviewer 2
Comments and Suggestions for Authors
The manuscript is interesting as a prospective study on A. platensis research, which takes on a linear form under laboratory conditions. Several aspects of morphology in relation to certain cultivation conditions have been addressed, including the sodium content from different sources. It is a starting point.
Observations:
The description of the experiments should be more detailed. Specifically, the linear form was isolated based on the cultivation of the Arthrospira platensis KCTC AG40101 strain on a medium containing... Was the inoculum for the experiments cultivated in the same medium?
Sol) Linear A. platensis was found in a culture where the coiled wild-type A. platensis was growing and was isolated through a selection process using filtering. Both strains were grown in the original SOT medium to compare their growth rate, har-vesting efficiency, and photosynthetic performance: 16.8 g/L NaHCO3, 0.5 g/L K2HPO4, 2.5 g/L NaNO3, 1.0 g/L K2SO4, 1.0 g/L NaCl, 0.2 g/L MgSO47H2O, 0.04 g/L CaCl22H2O, 0.01g/L FeSO47H2O, 0.07 g/L Na2EDTA2H2O, and 1 mL of a micronutrient solution. The micronutrient solution consisted of 2.86 g/L H3BO3, 1.81 g/L MnCl24H2O, 0.222 g/L ZnSO44H2O, 0.021 g/L Na2MoO42H2O, 0.08 g/L CuSO4 5H2O, and 0.05 g/L Co(NO3)26H2O.
The revised parts are highlighted in red color in the text of this manuscript (Line 79 - 85 on Page 2).
The addition of Na₂CO₃ and NaCl requires further justification. Na₂CO₃ is more basic than NaHCO₃, so the cultivation medium's pH should be mentioned. NaCl does not alter the pH but increases the Cl⁻ content.
Sol) We appreciate your comment. We added a graph showing pH changes under different culture conditions (Fig. 1, 2, and 3). We revised the corresponding figures and added the sentence about pH value.
The cultures using 0.1 M NaHCO₃, 0.2 M NaHCO₃, and 0.4 M NaHCO₃ maintained relatively constant pH values of 8.46, 8.66, and 8.91 on average during cultivation (Fig. 1B)
The revised parts are highlighted in red color in the text of this manuscript (Line 153 - 154 on Page 4).
Throughout the cultivation period, the addition of 0.2 M NaCl did not cause noticeable pH changes (pH 8.66) compared to the control condition (Fig. 2b).
The revised parts are highlighted in red color in the text of this manuscript (Line 194 - 195 on Page 5).
The initial pH of the culture condition using 0.2 M Na₂CO₃ was high at 10.02. However, due to aeration with 3% carbon dioxide, the average pH during the cultivation period was 8.92, which was not significantly higher than that of the control medium condition using 0.2 M NaHCO₃ (Fig. 3B).
The revised parts are highlighted in red color in the text of this manuscript (Line 216 - 220 on Page 5 - 6).
Results of practical importance, such as auto flotation and filtration, are included. The conclusion corresponds mainly to the work but is more of a summary of the experimental observations. An in-depth analysis of the industrial implications of these results would be welcome.
Sol) An overview of the current harvesting methods for A. platensis in both laboratory and industrial settings was provided in the text of this manuscript. Additionally, the auto-flotation harvesting method was compared with other harvesting methods in terms of efficiency, feasibility, and cost.
Refer the following sentences.
“Current harvesting methods for Spirulina (A. platensis) vary based on the scale of pro-duction. For laboratory-scale applications, common methods include filtration using membrane filters or mesh sieves [26,27], low-speed centrifugation [28], and sedimentation with or without flocculants [29,30]. In industrial-scale operations, more advanced tech-niques such as microfiltration [31], industrial centrifugation [32], flocculation-assisted sedimentation [33], belt and rotary drum filtration [34,35], and electrocoagulation [36,37] are employed to efficiently separate biomass while ensuring cost-effectiveness. The choice of harvesting method depends on factors such as scalability, energy consumption, and processing efficiency. Among these methods, auto-flotation offers significant advantages for harvesting Spirulina, as it is a low-cost, energy-efficient, and easily implementable method [38] compared to centrifugation, which is highly effective but energy-intensive [39]; filtration, which requires frequent maintenance due to clogging [40]; and floccula-tion-assisted sedimentation, which, while cost-effective, requires additional processing to remove residual flocculants [41].”
The revised parts are highlighted in red color in the text of this manuscript (Line 51 - 64 on Page 2).
[Reference]
[26] Ismail, I.; Kurnia, K.A.; Samsuri, S.; Bilad, M.R.; Marbelia, L.; Ismail, N.M.; Khan, A.L.; Budiman, A.; Susilawati, S. Energy efficient harvesting of Spirulina sp. from the growth medium using a tilted panel membrane filtration. Bioresource Technology Reports 2021, 15, 100697.
[27] Cuellar-Bermudez, S.P.; Kilimtzidi, E.; Devaere, J.; Goiris, K.; Gonzalez-Fernandez, C.; Wattiez, R.; Muylaert, K. Harvesting of Arthrospira platensis with helicoidal and straight trichomes using filtration and centrifugation. Separation Science and Technology 2020, 55, 2381-2390.
[28] Yong, T.C.; Chiu, P.-H.; Chen, C.-H.; Hung, C.-H.; Chen, C.-N.N. Disruption of thin-and thick-wall microalgae using high pressure gases: Effects of gas species, pressure and treatment duration on the extraction of proteins and carotenoids. Journal of bioscience and bioengineering 2020, 129, 502-507.
[29] Chatsungnoen, T.; Chisti, Y. Harvesting microalgae by flocculation–sedimentation. Algal Research 2016, 13, 271-283.
[30] Leite, L.d.S.; Daniel, L.A. Optimization of microalgae harvesting by sedimentation induced by high pH. Water Science and Technology 2020, 82, 1227-1236.
[31] Melanie, H.; Aspiyanto, A.; Maryati, Y.; Susilowati, A.; Mulyani, H.; Filailla, E.; Devi, A.F.; Budiari, S. Recovery of phycocyanin from microalgae Spirulina using membrane technology and evaluation of its antioxidant activity. In Proceedings of the AIP Conference Proceedings, 2023.
[32] Zhu, J.; Wakisaka, M.; Omura, T.; Yang, Z.; Yin, Y.; Fang, W. Advances in industrial harvesting techniques for edible microalgae: Recent insights into sustainable, efficient methods and future directions. Journal of Cleaner Production 2024, 140626.
[33] Liu, Z.; Hao, N.; Hou, Y.; Wang, Q.; Liu, Q.; Yan, S.; Chen, F.; Zhao, L. Technologies for harvesting the microalgae for industrial applications: Current trends and perspectives. Bioresource Technology 2023, 129631.
[34] Silva, N.C.; Machado, M.V.; Brandão, R.J.; Duarte, C.R.; Barrozo, M.A. Dehydration of microalgae Spirulina platensis in a rotary drum with inert bed. Powder Technology 2019, 351, 178-185.
[35] Silva, N.C.; Duarte, C.R.; Barrozo, M.A. Analysis of the use of a non-conventional rotary drum for dehydration of microalga Spirulina platensis. Bioprocess and biosystems engineering 2020, 43, 1359-1367.
[36] de Souza Sossella, F.; Rempel, A.; Nunes, J.M.A.; Biolchi, G.; Migliavaca, R.; Antunes, A.C.F.; Costa, J.A.V.; Hemkemeier, M.; Colla, L.M. Effects of harvesting Spirulina platensis biomass using coagulants and electrocoagulation–flotation on enzymatic hydrolysis. Bioresource Technology 2020, 311, 123526.
[37] Krishnamoorthy, N.; Unpaprom, Y.; Ramaraj, R.; Maniam, G.P.; Govindan, N.; Arunachalam, T.; Paramasivan, B. Recent advances and future prospects of electrochemical processes for microalgae harvesting. Journal of Environmental Chemical Engineering 2021, 9, 105875.
[38] Kim, S.G.; Choi, A.; Ahn, C.Y.; Park, C.S.; Park, Y.H.; Oh, H.M. Harvesting of Spirulina platensis by cellular flotation and growth stage determination. Letters in applied microbiology 2005, 40, 190-194.
[39] Najjar, Y.; Abu-Shamleh, A. Harvesting of microalgae by centrifugation for biodiesel production: a review. Algal Res 51: 102046. 2020.
[40] Mkpuma, V.O.; Moheimani, N.R.; Ennaceri, H. Microalgal dewatering with focus on filtration and antifouling strategies: A review. Algal Research 2022, 61, 102588.
[41] Vermuë, M.; Eppink, M.; Wijffels, R.; Van Den Berg, C. Multi-product microalgae biorefineries: from concept towards reality. Trends in biotechnology 2018, 36, 216-227
The experimental results come from controlled environments, which may not fully reflect the behavior of the filaments in more complex natural environments. The conclusion suggests spiral filaments dominate in natural environments due to their higher buoyancy. Still, the paper did not detail the competitive advantages of this morphology in terms of survival and ecological adaptation, as it had a different objective.
Sol) I agree with your option. The two different types of Spirulina were cultivated under limited conditions in a controlled environment, such as a laboratory, and therefore cannot fully simulate the complexities of natural environmental conditions.
Refer the following sentence.
“However, this assumption is based on results obtained in a controlled environment and may not fully reflect Spirulina's behavior in more complex natural conditions.”
The revised parts are highlighted in red color in the text of this manuscript (Line 364 - 366 on Page 11).
We detailed the competitive advantages of this morphology in terms of biomass accumulation and harvesting efficiency.
Refer the following sentence.
Due to its distinct morphology, linear Spirulina achieves higher photosynthetic efficiency, enabling greater biomass accumulation. However, it exhibits significantly lower harvesting efficiency through filtration or auto-flotation. In the commercialization of linear and coiled Spirulina, the benefits of enhanced harvesting efficiency in the coiled form significantly outweigh the biomass accumulation losses compared to the linear type.
The revised parts are highlighted in red color in the text of this manuscript (Line 366 - 371 on Page 11).
The study aimed to compare the growth rate, photosynthetic activity, and auto-flotation efficiency of the linear and spiral types of A. platensis. However, the analyzed article does not provide concrete and explicit information about the exact medium in which the spiral forms of A. platensis were cultivated. Were they subjected to the same sodium concentrations? Although comparisons between linear and spiral forms exist, the methodology for spiral forms is not described in detail, and the specific medium in which they were grown is not directly mentioned.
Sol) Both strains were grown in the original SOT medium to compare their growth rate, harvesting efficiency, and photosynthetic performance: 16.8 g/L NaHCO3, 0.5 g/L K2HPO4, 2.5 g/L NaNO3, 1.0 g/L K2SO4, 1.0 g/L NaCl, 0.2 g/L MgSO47H2O, 0.04 g/L CaCl22H2O, 0.01g/L FeSO47H2O, 0.07 g/L Na2EDTA2H2O, and 1 mL of a micronutrient solution. The micronutrient solution consisted of 2.86 g/L H3BO3, 1.81 g/L MnCl24H2O, 0.222 g/L ZnSO44H2O, 0.021 g/L Na2MoO42H2O, 0.08 g/L CuSO4 5H2O, and 0.05 g/L Co(NO3)26H2O.
The revised parts are highlighted in red color in the text of this manuscript (Line 79 - 85 on Page 2).
Linear Spirulina was cultured under the same medium composition and cultivation conditions as coiled Spirulina, ensuring that the medium composition, including sodium content, remained identical.
The spiral and linear forms were compared in terms of growth rate, photosynthetic activity, and buoyancy. This suggests that the spiral forms were partially grown in similar media to provide a comparative basis.
Many questions arise; therefore, a detailed description of the experimental design would be beneficial.
Sol) Both strains were grown in the original SOT medium to compare their growth rate, harvesting efficiency, and photosynthetic performance.
The revised parts are highlighted in red color in the text of this manuscript (Line 79 - 80 on Page 2).
The statement that the difference in buoyancy may explain the rarity of linear filaments in natural environments indirectly implies that the morphology of A. platensis filaments (linear or coiled) is reversible or influenced by environmental conditions. However, the article does not explicitly mention the reversibility of linear and coiled forms. The linear form was obtained by spontaneously converting coiled filaments under laboratory conditions. However, the paper does not clarify whether this change is permanent or if linear filaments can revert to the coiled form under natural conditions.
Sol) Thank you for your comment. The linear filaments of A. platensis in our study have not reverted to their original coiled morphology. We incorporated additional relevant experiments into the Materials and Methods section and included the corresponding results in the main text.
The revised parts are highlighted in red color in the text of this manuscript (Line 105 - 108 on Page 3).
The revised parts are highlighted in red color in the text of this manuscript (Line 339 - 341 on Page 10).
The differences between the two forms are discussed but without details regarding their reversibility or how these forms coexist or transform in nature.
Sol) After screening linear Spirulina, no morphological reversion from the linear to the coiled form has been observed to date. The two different types of Spirulina were cultivated under controlled environment, such as a laboratory. Due to the limitations of our experimental conditions, fully simulating the complexity of natural environmental conditions is not feasible. Please take into account the limitations when interpreting the reversibility and coexistence of the two strains in natural ecosystems.
It is suggested that linear filaments' lower buoyancy makes them less competitive in natural environments, but no experimental data is provided to demonstrate whether linear filaments obtained in the laboratory can revert to the coiled form or vice versa.
Sol) To determine whether the selected linear filaments could revert to their original coiled morphology, all filament samples were observed under a microscope (OLYMPUS CKX41, Japan) during subculturing for one year.
The revised parts are highlighted in red color in the text of this manuscript (Line 105 - 108 on Page 3).
In our study, the linear filaments of A. platensis have not reverted to their original coiled morphology to date.
Therefore, we added the result to the main text.
“Moreover, in our experiment, following this screening procedure, we observed no morphological reversion from the linear to coiled form, although linear filaments were readily generated from coiled filaments.”
The revised parts are highlighted in red color in the text of this manuscript (Line 339 - 341 on Page 10).
From this arises the question: Is the A. platensis KCTC AG40101 strain maintained on the described mineral medium?
Sol) Linear Spirulina was cultured under the same medium composition and cultivation conditions as coiled Spirulina. The medium composition for linear Spirulina was described in the section 2.1.
The revised parts are highlighted in red color in the text of this manuscript (Line 79 - 85 on Page 2).
Regarding the figures, statistical analysis is not described (information about the number of independent experiments and data processing is missing from the Methods section).
Sol) Thank you for the suggestion. We added statistical indications to Figures 1, 2, 3, 4, 5, 7, 8, and 9 using SAS ver. 9.4 (SAS Institute Inc., Cary, NC, USA) to enhance the reliability and interpretation of the data. Data were analyzed using one-way ANOVA followed by Duncan’s test (p <0.05).
The abbreviations (Figures 7 and 8) are clear in that they refer to the form of A. platensis, but this should be explicitly mentioned.
Sol) The abbreviations in Figures 7 and 8 have been clarified to explicitly indicate the form of A. platensis.
"CSP" has been revised to "Coiled filament," and "LSP" has been revised to "Linear filament."
Figures 7 and 8 was replaced with revised versions.

Reviewer 3 Report
Comments and Suggestions for Authors
Comments on microorganisms-3451986 entitled “Relationship between harvesting efficiency and filament morphology in Arthrospira platensis”
The study is devoted to the analysis of the relationship of different type’s filament length of Arthrospira platensis Gomont (Oscillatoriales, Cyanobacteria) with concentrations of sodium compounds in the growth medium, and also the effect of filament morphology on the harvest efficiency using filtration and auto-flotation. A. platensis is widely used in biotechnology. However, the relationship of harvest efficiency with morphological conversion in different strains of this species has not been sufficiently analyzed. Due to this current research is relevant.
The study is well presented, but needs some improvement.
I have some remarks dealing with this manuscript:
1) The Title, line 3. The authors of the species are not listed at first mention in the text. “Arthrospira platensis” should be corrected to “Arthrospira platensis Gomont”.
2) Keywords, line 22. “Arthrospira platensis” should be corrected to “Arthrospira platensis”.
3) Line 26. Cyanophyceae is not a phylum, but a class (Guiry, M.D.; Guiry, G.M. AlgaeBase. World-Wide Electronic Publication, National University of Ireland, Galway. 2025. Available online: https://www.algaebase.org (accessed on 21 January 2025). Therefore, the word "phylum" should be corrected to "class".
4) Lines 37-39. Perhaps the authors should consult the paper by Yadav et al. (2019), which analyzed genomic changes in two morphotypes with only helical or only straight trichomes of one strain of Arthrospira, to complement their paper (Yadav, A., Monsieurs, P., Misztak, A., Waleron, K., Leys, N., Cuypers, A., & Janssen, P. J. (2019). Helical and linear morphotypes of Arthrospira sp. PCC 8005 display genomic differences and respond differently to 60Co gamma irradiation. European Journal of Phycology, 55(2), 129–146. https://doi.org/10.1080/09670262.2019.1675763).
5) Lines 78-79. “Subsequently, we observed changes in filament morphology over time under different conditions” should be corrected to “Subsequently, we observed changes in filament morphology over time under different conditions.”.
6) Lines 84, 87-88. The manufacturer and country of the devices should be given.
7) Lines 84-85. It is not clear what statistical methods the authors used. Are the results obtained by measuring only ten filaments reliable?
8) Line 153. “…A. Platensis…” should be corrected to “…A. platensis…”.
9) Lines 154, 156-157, 297-298. References should be given correctly.
10) Line 281. “…Synechocystis Olive…” should be corrected to “…Synechocystis Sauvageau…”.
Author Response
Reviewer 3
Comments on microorganisms-3451986 entitled “Relationship between harvesting efficiency and filament morphology in Arthrospira platensis”
The study is devoted to the analysis of the relationship of different type’s filament length of Arthrospira platensis Gomont (Oscillatoriales, Cyanobacteria) with concentrations of sodium compounds in the growth medium, and also the effect of filament morphology on the harvest efficiency using filtration and auto-flotation. A. platensis is widely used in biotechnology. However, the relationship of harvest efficiency with morphological conversion in different strains of this species has not been sufficiently analyzed. Due to this current research is relevant.
The study is well presented, but needs some improvement.
I have some remarks dealing with this manuscript:
- The Title, line 3. The authors of the species are not listed at first mention in the text. “Arthrospira platensis” should be corrected to “Arthrospira platensisGomont”.
Sol) The term ' Arthrospira platensis ' was revised to ' Arthrospira platensis Gomont ' to ensure proper scientific nomenclature and adherence to taxonomic conventions. (see Line 3)
The revised parts are highlighted in red color in the text of this manuscript (Line 3 on Page 1).
- Keywords, line 22. “Arthrospira platensis” should be corrected to “Arthrospira platensis”.
Sol) The term Arthrospira platensis in the keywords has been revised to italicize the species name, ensuring adherence to scientific formatting conventions (see Line 22).
The revised parts are highlighted in red color in the text of this manuscript (Line 22 on Page 1).
- Line 26. Cyanophyceae is not a phylum, but a class (Guiry, M.D.; Guiry, G.M. AlgaeBase. World-Wide Electronic Publication, National University of Ireland, Galway. 2025. Available online: https://www.algaebase.org (accessed on 21 January 2025). Therefore, the word "phylum" should be corrected to "class".
Sol) We appreciate your comments. The term "phylum" was revised to "class" to accurately reflect the taxonomic rank of Cyanophyceae, as referenced in Guiry and Guiry (2025) (see Line 26).
The revised parts are highlighted in red color in the text of this manuscript (Line 26 on Page 1).
- Lines 37-39. Perhaps the authors should consult the paper by Yadav et al. (2019), which analyzed genomic changes in two morphotypes with only helical or only straight trichomes of one strain of Arthrospira, to complement their paper (Yadav, A., Monsieurs, P., Misztak, A., Waleron, K., Leys, N., Cuypers, A., & Janssen, P. J. (2019). Helical and linear morphotypes of Arthrospira PCC 8005 display genomic differences and respond differently to 60Co gamma irradiation. European Journal of Phycology, 55(2), 129–146. https://doi.org/10.1080/09670262.2019.1675763).
Sol) Thank you for the suggestion. We referred to the paper by Yadav et al. (2019) and added the relevant sentence in section of Introduction.
“A previous study reported that genomic differences are observed between the helical and linear morphotypes of Arthrospira sp. PCC 8005 [19].”
Additionally, the reference has been included in the References section.
[Reference]
[19] Yadav, A.; Monsieurs, P.; Misztak, A.; Waleron, K.; Leys, N.; Cuypers, A.; Janssen, P.J. Helical and linear morphotypes of Arthrospira sp. PCC 8005 display genomic differences and respond differently to 60Co gamma irradiation. European journal of phycology 2020, 55, 129-146.
The revised parts are highlighted in red color in the text of this manuscript (Line 39 - 40 on Page 1)
- Lines 78-79. “Subsequently, we observed changes in filament morphology over time under different conditions” should be corrected to “Subsequently, we observed changes in filament morphology over time under different conditions.”
Sol) A period was added at the end of the sentence to correct the punctuation (see Lines 80–81).
“Subsequently, we observed changes in filament morphology over time under different conditions.”
The revised parts are highlighted in red color in the text of this manuscript (Line 94 - 95 on Page 2)
- Lines 84, 87-88. The manufacturer and country of the devices should be given.
Sol) The manufacturer and country of the devices have been added for clarity.
“OLYMPUS CKX41” was revised to “OLYMPUS CKX41, Japan”
The revised parts are highlighted in red color in the text of this manuscript (Line 107, 113 on Page 3)
“Whatman” was revised to “Whatman, USA”
The revised parts are highlighted in red color in the text of this manuscript (Line 117 on Page 3)
- Lines 84-85. It is not clear what statistical methods the authors used. Are the results obtained by measuring only ten filaments reliable?
Sol) We added statistical indications in Figures 1c, 2c, and 3c using SAS ver. 9.4 (SAS Institute Inc., Cary, NC, USA) to enhance the reliability and interpretation of the data.
- Line 153. “… Platensis…” should be corrected to “…A. platensis…”.
Sol) The capitalization of “A. Platensis” has been corrected to “A. platensis” to adhere to proper scientific naming conventions (see Line 185).
The revised parts are highlighted in red color in the text of this manuscript (Line 185 on Page 4)
- Lines 154, 156-157, 297-298. References should be given correctly.
Sol) Thank you for the suggestion. The reference style has been revised for consistency and alignment with the journal’s guidelines.
About Line 154
“For example, Kebede (1997) observed abnormally long trichomes under low salinity condi-tions (13 g/L) due to salinity stress in the presence of different sodium salts (NaHCO3, NaCl, Na2SO4), but short and dense spiral structures in high-salinity NaCl-based media (55–68 g/L)”
The above sentence has been revised as follows:
“Low salinity conditions (13 g/L) in the presence of different sodium salts (NaHCO3, NaCl, Na2SO4) led to the observation of abnormally long trichomes due to salinity stress, whereas high-salinity NaCl-based media (55–68 g/L) resulted in short and dense spiral structures [26]”
Additionally, the reference has been included in the References section (see Lines 185–188).
The revised parts are highlighted in red color in the text of this manuscript (Line 185 - 188 on Page 5)
About Line 156-157
“Nosratimovafagh et al. (2023) demonstrated that helix length, diameter, and pitch length were affected by the light spectrum, NaCl concentration, and glucose level.”
The above sentence has been revised as follows:
“Additionally, another study demonstrated that helix length, diameter, and pitch length were affected by the light spectrum, NaCl concentration, and glucose level [27]”
Additionally, the reference has been included in the References section (see Lines 188–190).
The revised parts are highlighted in red color in the text of this manuscript (Line 188 - 190 on Page 5)
About Line 297-298
“The morphological conversion of coiled filaments to linear filaments was mainly be-lieved to be heritable and stable due to genetic drift (Wang and Zhao, 2013, Muhling et al. 2013).”
The above sentence has been revised as follows:
“The morphological conversion of coiled filaments to linear filaments was mainly be-lieved to be heritable and stable due to genetic drift [25,56]”
Additionally, the reference has been included in the References section (see Lines 332–333).
The revised parts are highlighted in red color in the text of this manuscript (Line 332 - 333 on Page 10)
- Line 281. “…Synechocystis Olive…” should be corrected to “…Synechocystis Sauvageau…”.
Sol) Synechocystis Olive is antenna mutant which lack PC subunit. The strain allow higher light penetration into the reactor volume because antenna deficiency reduces self-shading. As a result, the Olive strain exhibits a higher growth rate than Synechocystis WT due to increased photosynthetic efficiency. Synechocystis Olive is the strain we use, and its designation is considered appropriate.
[Reference]
[50] Kwon, J.-H.; Bernat, G.; Wagner, H.; Roegner, M.; Rexroth, S. Reduced light-harvesting antenna: consequences on cyanobacterial metabolism and photosynthetic productivity. Algal Research 2013, 2, 188-195.

Round 2
Reviewer 1 Report
Comments and Suggestions for Authors
The current manuscript is a better version after revision. However, here are still some issues needing further confirmation and improvement:
1. I still recommend to mention that (1) increasing concentration of NaHCO3 could increase the filament length of A. platensis (seen from Fig. 1c), (2) increasing concentration of NaCl could increase the filament length of A. platensis (seen from Fig. 2c). Because in the two experiments (shown by Fig. 1c and Fig. 2c), there was more than one variables except Na+ concentration.
2. Line161-162, did the author measure the NaHCO3 concentration in the medium along cultivation?
Author Response
Reviewer 1 (Round 2)
The current manuscript is a better version after revision. However, here are still some issues needing further confirmation and improvement:
- I still recommend to mention that (1) increasing concentration of NaHCO3 could increase the filament length of A. platensis (seen from Fig. 1c), (2) increasing concentration of NaCl could increase the filament length of A. platensis (seen from Fig. 2c). Because in the two experiments (shown by Fig. 1c and Fig. 2c), there was more than one variables except Na+ concentration.
Sol) We appreciate your comment.
We added the corresponding sentence in the text of this manuscript.
“These results show that the NaHCO3 concentration affected the morphology of A. platensis filaments, in that increasing concentration of NaHCO3 could increase the filament length of A. platensis (Fig. 1c).
The revised parts are highlighted in red color in the text of this manuscript (Line 164 - 165 on Page 4).
“Thus, these experiments confirmed that increasing concentration of NaCl could increase the filament length of A. platensis (Fig. 2c) but also inhibited cell growth, making this medium unsuitable for culturing of A. platensis.”
The revised parts are highlighted in red color in the text of this manuscript (Line 199 - 201 on Page 5).
- Line161-162, did the author measure the NaHCO3 concentration in the medium along cultivation?
Sol) We appreciate your comment. As Spirulina grows, the initial concentration of NaHCO₃ in the medium decreases. However, we did not measure the remaining NaHCO₃ concentration during the cultivation process. We initially used the term "depletion" in the text, but this was incorrect. Since 0.4 M NaHCO₃ was added at the beginning of the cultivation, NaHCO₃ depletion is unlikely to occur after 18 days of cultivation. Since the cultivation lasted for more than 18 days, we revised the text to suggest that the reduction in NaHCO₃ concentration in the medium, caused by the increase in Spirulina cell number, may explain the decrease in Spirulina filament length. Thank you for correcting our mistake.
We revised the corresponding sentence.
“This decrease in filament length after day 18 is likely attributable to reduction of NaHCO₃ in the growth medium as a result of cellular growth.”
The revised parts are highlighted in red color in the text of this manuscript (Line 161 - 163 on Page 4).

Reviewer 2 Report
Comments and Suggestions for Authors
The manuscript has been improved by clarifying experimental details and deepening the interpretation of the results, providing valuable data for researchers interested.
Author Response
Thank you for your valuable feedback. We appreciate your recognition of the improvements in our manuscript and hope that the provided data will be beneficial to researchers in the field.